# Storm time polar cap expansion: IMF clock angle dependence

Beket Tulegenov[1], Joachim Raeder[1], William D. Cramer[1], Banafsheh Ferdousi[1], Timothy J. Fuller-Rowell[2], Naomi Maruyama[3], and Robert J. Strangeway[4]

[1]Space Science Center, University of New Hampshire, Durham, NH, USA
[2]Cooperative Institute for Research in Environmental Sciences, University of Colorado, Boulder, CO, USA
[3]NOAA Space Weather Prediction Center, Boulder, CO, USA
[4]University of California, Los Angeles, CA, USA

**Correspondence:** J. Raeder (J.Raeder@unh.edu)

**Abstract.**

It is well known that the polar cap, delineated by the Open Closed field line Boundary (OCB), responds to changes in the Interplanetary Magnetic Field (IMF). In general, the boundary moves equatorward when the IMF turns southward and contracts poleward when the IMF turns northward. However, observations of the OCB are spotty and limited in local time, making more detailed studies of its IMF dependence difficult. Here, we simulate five solar storm periods with the coupled model consisting of the Open Geospace General Circulation model (OpenGGCM) coupled with Coupled Thermosphere Ionosphere Model (CTIM) and the Rice Convection Model (RCM), i.e., the OpenGGCM-CTIM-RCM model, to estimate the location and dynamics of the OCB. For these events, polar cap boundary location observations are also obtained from Defense-Meteorological Satellite Program (DMSP) precipitation spectrograms and compared with the model output. There is a large scatter in the DMSP observations and in the model output. Although the model does not predict the OCB with high fidelity for every observation, it does reproduce the general trend as a function of IMF clock angle. On average, the model overestimates the latitude of the open-closed field line boundary by 1.61 degrees. Additional analysis of the simulated polar cap boundary dynamics across all local times shows that the MLT of the largest polar cap expansion closely correlates with the IMF clock angle; that the strongest correlation occurs when the IMF is southward; that during strong southward IMF the polar cap shifts sunward; and that the polar cap rapidly contracts at all local times when the IMF turns northward.

## 1 Introduction

The total magnetic flux contained in the open magnetic field lines of Earth's polar caps is a crucial parameter for the energy stored in the magnetosphere, and thus for substorm and storm dynamics (Siscoe and Huang, 1985; Milan et al., 2008). The amount of the open flux is essentially given by the polar cap area, which is bounded by the OCB. The shape and dynamics of the OCB are ultimately controlled by magnetic reconnection and large-scale convection (Cowley, 1982; Milan et al., 2007). Magnetic flux enters and exits the polar cap through reconnection at the magnetopause and in the tail. However, convection can also change the shape of the OCB without changing the flux contained in the polar cap (Lockwood et al., 1990). Knowing the OCB location during strong solar events is crucial for modeling the ionosphere-thermosphere system, because the interaction

is different on open field lines than on closed field lines. Specifically, the location of Field Aligned Current (FAC) systems is determined by the OCB, as well as the regions of auroral precipitation. In particular, the strong electron precipitation that originates from the trapped particle populations of the inner magnetosphere abruptly cuts off at the OCB. This property is routinely used to determine the OCB from particle measurements on Low Earth Orbiting (LEO) satellites, and we adopted the same technique in this study. Precipitation also significantly impacts ionosphere-thermosphere dynamics through heating and conductance. Ion outflow from the ionosphere is also affected by the field topology. Outflow on open field lines eventually leaves the magnetosphere entirely through the lobes or becomes recaptured by tail reconnection. On the other hand, the outflow on closed field lines becomes trapped and forms the plasmasphere. When the polar cape opens up, that plasma leaves the plasmasphere and convects away. Thus, the OCB shape also controls the shape of the plasmasphere (Nishida, 2019).

The opening of the closed magnetic field lines on the dayside due to the incoming southwardly oriented IMF and the consequent reconnection of these field lines on the nightside is known as the Dungey cycle (Dungey, 1961; Siscoe and Huang, 1985; Milan et al., 2003). It is one of the main drivers of the magnetosphere and the ionosphere, besides viscous interaction at the magnetopause. The open flux balance between dayside and nightside reconnection rates controls the convection (Siscoe and Huang, 1985; Hubert et al., 2006).

Milan et al. (2007, 2012), using radar data, found that the rate of change of the polar cap area can be used to estimate open field flux and, in turn, measure dayside and nightside reconnection rates. It has also been determined that the IMF dynamics influences the location of the OCB and the response of the large-scale convection (Maynard, 2003; Lockwood et al., 2006). Milan et al. (2003) has combined various ground and space-based measurements to determine the polar cap area changes during two substorm cycles and they have shown that the polar cap increased in size when the IMF was southward oriented.

There are various methods of determining the location of the OCB using observations. From the ground, optical auroral emissions at 6300 Å are measured using meridian scanning photometers. The poleward boundary of these emissions is used as a proxy for OCB (Blanchard et al., 1995; Johnsen and Lorentzen, 2012) because these emissions are from precipitating electrons that come from the closed field lines of the inner magnetosphere. Another method for OCB detection uses HF radar networks such as EISCAT and SuperDARN (Pinnock and Rodger, 2000; Aikio et al., 2006, 2013), which essentially also observe precipitation effects. These methods provide reasonable OCB latitude estimates when the polar cap does not expand beyond its regular steady-state size. During times of high geomagnetic activity these methods can fail because the precipitation is very intense, clobbering the radars' return signal (Gauld et al., 2002; Gillies et al., 2011).

Spaceborne optical measurements can provide global coverage during dynamic events. By using the poleward boundary of the auroral ultraviolet emissions the location of the OCB can be identified. Studies have shown that Ultraviolet Imager (UVI) on board of Polar spacecraft can be used for these purposes (Elsen et al., 1998; Milan et al., 2003). However, these measurements are also of limited value during disturbed times, when a clear OCB often cannot be identified from the data. The DMSP satellite measurements provide an alternative way for the estimation of OCB latitudes. The constellation of three to four spacecraft provides electron precipitation measurements. The poleward boundary of precipitating particles with energies up to 10s of keV can often be clearly identified in the spectra and can be used to identify the polar cap boundary (Sotirelis et al.,

1998; Sotirelis, 2005; Wing and Zhang, 2015; Wang et al., 2018). However, one of the drawbacks of using satellites for OCB mapping is a lack of simultaneous coverage for all, or even just more than a few, local times.

Early global MHD simulations have shown that OCB locations for all local times could be estimated using numerical methods (Raeder et al., 1998; Lopez et al., 1999). Parametric studies based on the MHD models have shown that the location of the OCB changes when the IMF clock angle changes (Kabin, 2004; Wang et al., 2016). Comparison between MHD model outputs and the satellite data for specific steady-state events has shown a good agreement between OCB latitudes from the model and the observations (Rae, 2004). Analysis of the Bastille Day CME based on the IMAGE and Polar satellites showed

that MHD models could be used to estimate the global shape and location of the polar cap (Rastätter, 2005). However, models used at the time were not successful in reproducing the small-scale arcs along the polar cap boundary. Rae et al. (2010) have shown the importance of the inclusion of the ring current models in the MHD simulations when determining the OCB location. The more recent study by Wang et al. (2018) has presented a comparison between the DMSP OCB and the PPMLR-MHD model output for a single substorm event. The difference between the model and the satellite OCB latitudes was $2.33°$ on

average. The model, however, used a fixed $B_x$ component of the IMF, empirical conductance models, and did not incorporate any ring current model.

    In this study we employ the OpenGGCM-CTIM-RCM coupled model to estimate the size, location, and dynamics of the polar cap during strong solar events. Our main finding, previously undocumented, is that the shape of the polar cap is closely correlated with the IMF clock angle.

## 75  2   Methodology

### 2.1   Model

This study is based on simulations performed using OpenGGCM-CTIM global 3D magnetohydrodynamic (MHD) magnetosphere model (Fuller-Rowell et al., 1996; Raeder et al., 2001, 2008), coupled with the Rice Convection Model (RCM) that models the ring current (Toffoletto et al., 2003). Details of the coupling methodology between these models were presented by

80 Cramer et al. (2017).

    The 3D magnetospheric modeling domain is defined using a geocentric solar ecliptic (GSE) coordinate system. It extends 35 $R_E$ upstream into the solar wind and to 5000 $R_E$ downstream along the X-axis. The domain spans $\pm48$ $R_E$ along both the Y and Z-axis. The numerical grid has a stretched Cartesian topology that provides a high resolution near the Earth and lower resolution elsewhere to minimize computational cost, while providing sufficient resolution in the region of interest. In this

study we use a grid with 481 grid points in the X direction, and 180 grid points in both the Y and Z directions. The RCM grid is defined in the ionosphere and encompasses only regions of closed magnetic field lines. It forms a belt around the Northern Hemisphere in a solar magnetic (SM) coordinate system from $45°$ to $82°$ in latitude. However, the actual RCM domain varies and adapts to the open-closed boundary, such that it only covers regions on closed field lines (Toffoletto et al., 2003). In the simulations presented here, RCM has 300 grid points in latitude and 101 grid points in the azimuthal direction. The grid is

non-uniform in latitude, and is denser at higher latitudes.

**Table 1.** List of the modeled storms with the mean and standard deviation of the differences in OCB latitude between the DMSP and OpenGGCM results. The second to last column is the mean model bias, where positive values indicate that the model on average predicts a lower latitude of the OCB, and the last column is the variance of the difference between the data and the model prediction.

| Start time (UT) | End time (UT) | Driver type | Min. Dst [nT] | $\overline{\Delta\theta}$ | $\delta\Delta\theta$ |
|---|---|---|---|---|---|
| 17:30 19/11/2003 | 17:30 21/11/2003 | CME | -422 | -0.30° | 4.57° |
| 07:00 30/09/2012 | 13:00 01/10/2012 | CME | -108 | 0.88° | 3.57° |
| 08:00 31/10/2012 | 04:00 02/11/2012 | CME | -65 | 1.11° | 3.23° |
| 18:00 12/11/2012 | 12:15 14/11/2012 | CME | -122 | 2.31° | 3.24° |
| 10:00 09/03/2018 | 00:00 11/03/2018 | CIR | -39 | 4.12° | 2.20° |

The coupled numerical model is driven by observed solar wind and IMF data at Lagrangian 1 (L1). The solar wind and IMF parameters like magnetic field, velocity, pressure, temperature, and density are obtained from the OMNIWeb. This dataset contains data that are obtained from L1 observations and shifted ballistically to 30 $R_E$ upstream of Earth (King, 2005). Since that is also the location of the model inflow boundary, no additional time shift is necessary. The solar F10.7 flux is used as a proxy for solar UV/EUV radiation in CTIM. The sunspot number is required for charge exchange calculations in the RCM model.

## 2.2 Event selection

In order to determine the behavior of the polar cap under dynamic solar storm conditions, four Coronal Mass Ejection (CME) events, and one Corotating Interaction Region (CIR) event were identified in the period from 2003 to 2019. The main selection criteria were dictated by the availability of solar wind and IMF data, as well as good coverage with DMSP data for the comparisons. All events have minimum Dst values below -35 nT. We also required the $B_y$ and $B_z$ components of the IMF to be larger than 10 nT in magnitude in the GSE coordinate system. These five solar storm periods are listed in Table 1. In addition we considered three more cases, where we changed the signs of IMF $B_y$, $B_z$, or both, in event 1, to test a hypothesis which is described in detail in Section 3.3.

## 3 Results

### 3.1 Comparison with DMSP data

Before we consider a detailed analysis of the simulation runs, we first assess the realism of the simulations by comparing the model OCB latitude output with DMSP observations. The DMSP satellites are a series of polar-orbiting spacecraft with an altitude of $\approx 850$ km in sun-synchronous orbits. The precipitating ion and electron data from the on-board instruments have been used to identify polar cap boundary crossings in a number of previous studies (Hardy et al., 1984; Sotirelis et al., 1998; Milan et al., 2003; Wing and Zhang, 2015; Wang et al., 2016, 2018). Spectrograms of ion and electron differential fluxes in a

range from 30 eV to 30 keV were inspected to identify the polar cap boundary crossings of the satellites. Such visual inspection is subjective. However, crossings that could not be clearly identified are not included in the database. Also, when we could identify polar cap precipitation features such as polar cap arcs, these were not included. We also never used data below 1 keV, so there should be no concerns about the cusp. All crossings are tabulated in the database (to be found in the supplemental material) with a time tag, so readers can double-check our boundary identifications.

Some previous studies have suggested using the b6 boundary as an open-closed field boundary (Newell et al., 1996; Hubert et al., 2006). These boundary locations can also be identified using a set of quantitative algorithms developed by Newell et al. (1996). However, during geomagnetically active periods these algorithms tend to either fail to determine the boundary at all, or sometimes they misidentify the boundary.

The comparison of the OCB magnetic latitudes between DMSP and the OpenGGCM simulations is shown in Fig. 1 through 5. Figure 1 shows the results for the November 20, 2003 event. Figure 1d shows the IMF and solar wind for reference. For this event, data was available from four DMSP spacecraft (F13-F16). The blue markers in Fig. 1a indicate the polar cap crossing latitudes of the DMSP satellites, while the modeled OCB latitudes, determined along the same orbits, are shown in red. Figure 1b displays the Magnetic Local Time (MLT) coverage of the DMSP spacecraft during the event. The DMSP s/c are in sun-synchronous orbits that are close to the terminator. Thus, the MLT coverage is uneven and concentrated in the dawn and dusk sectors. Near noon data are sparse, and there are no nightside crossings between 2200 and 0600 MLT. Figure 1a shows that the model follows the real OCB pattern geometry reasonably well. However, there is significant scatter, both in the data, as well as in the model results. In particular, the DMSP crossings often jump considerably from one orbit to the next, and the model shows a similar behavior. Obviously, the OCB is quite dynamic.

Figure 1c shows a histogram of the differences between the model and the DMSP OCB determination. The visual method of determining precipitation boundary from the DMSP spectrograms is fairly accurate, probably better than one degree. However, this method assumes that the precipitation boundary is also the OCB. During very active solar times that may not always be the case. There is a significant number of polar cap crossings where no clear precipitation boundary can be identified. Such cases are not included in the plots, but their existence indicates that the OCB and precipitation boundary may not always be the same. The simulation results shown in Fig. 1 through 7 also show that the OCB in the model is very dynamic and not always a smooth curve, but rather corrugated. That would also explain some of the scatter, assuming that the OCB in nature behaves similarly. The comparison shown in Fig. 1c indicates that on average the difference between DMSP and model OCB latitudes is $-0.30 \pm 4.57°$. Thus, in spite of the scatter, there is no significant bias between the model and the data for this event case. This, together with the fact that the data and the model pattern follow each other, this gives us confidence that the model results represent the true OCB to a high degree.

The other storm periods have data available from only three DMSP satellites (F16-F18). Figure 2 displays the OCB comparison for the October 1, 2012 event, and the histogram in Fig. 2c shows that the difference in OCB latitudes is $0.88 \pm 3.57°$. The comparison for the November 1, 2012 event shown in Fig. 3c gives a difference of $1.11 \pm 3.23°$ in the mean OCB latitude. The histogram for the November 13, 2012 event shown in Fig. 4c provides a mean difference of $2.31 \pm 3.24°$. Finally, the March

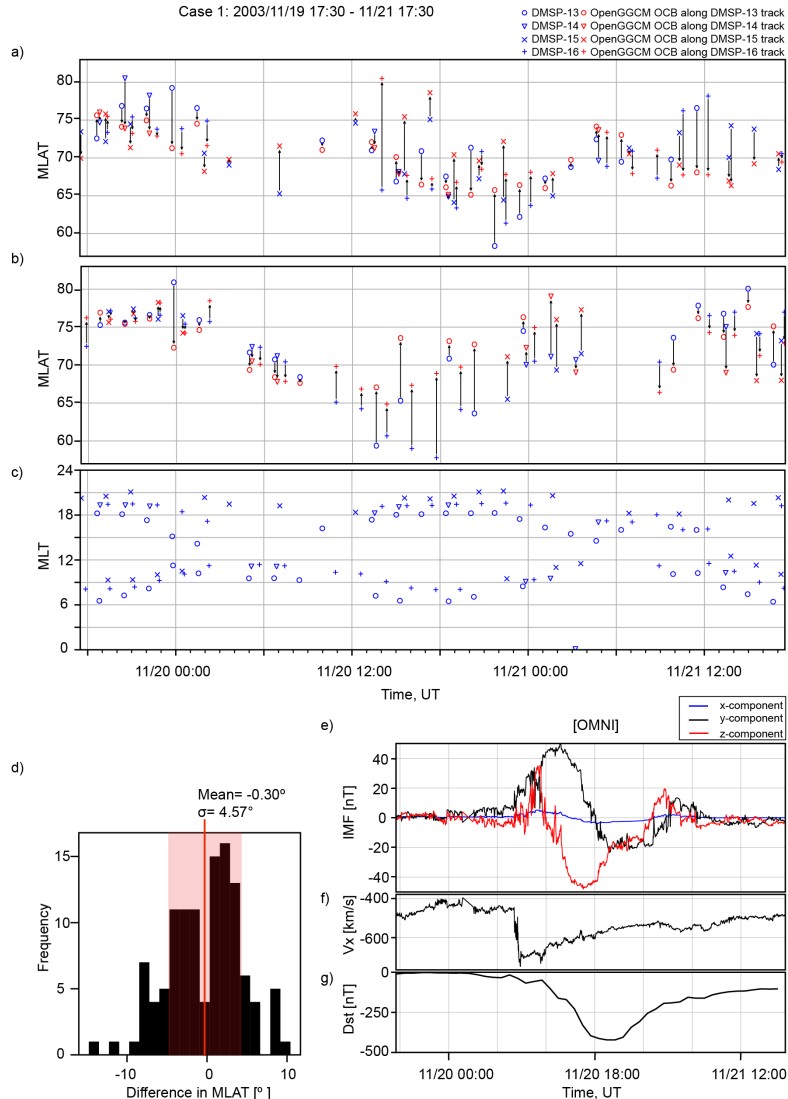

**Figure 1.** Comparison between the observed and simulated OCB latitudes for November 20, 2003 event. Panel a) shows the DMSP based OCB latitude in blue and corresponding OpenGGCM OCB latitude in red for all passes of the northern polar cap. Panel b) shows the DMSP based OCB latitude in blue and corresponding OpenGGCM OCB latitude in red for all passes of the southern polar cap. Panel c) indicates the coverage of the DMSP satellites for this event. A histogram of the differences between the observed and simulated OCB latitudes is shown in panel d). Panels e) and f) show the solar wind and IMF data used as input for the simulations. Panel g) shows the Dst index. The solar wind and IMF values are in GSE coordinates and were obtained from OMNIWeb.

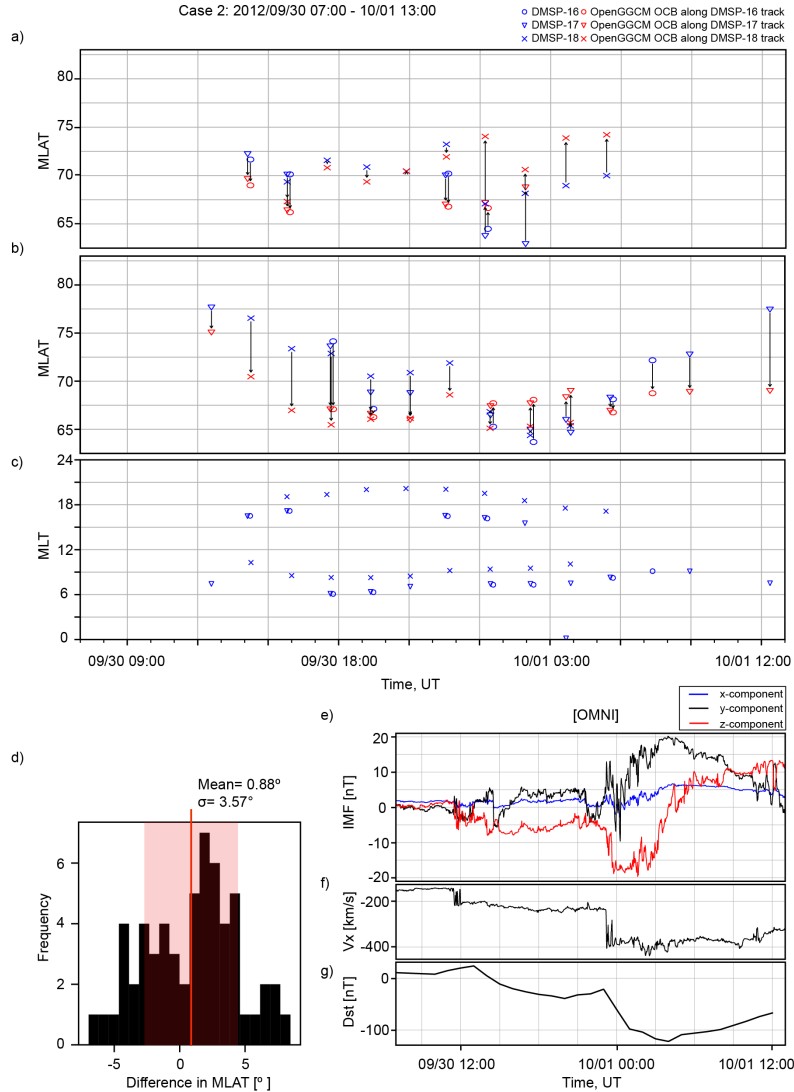

**Figure 2.** Comparison between the observed and simulated OCB latitudes for the October 1, 2012 event in the same format as Fig. 1.

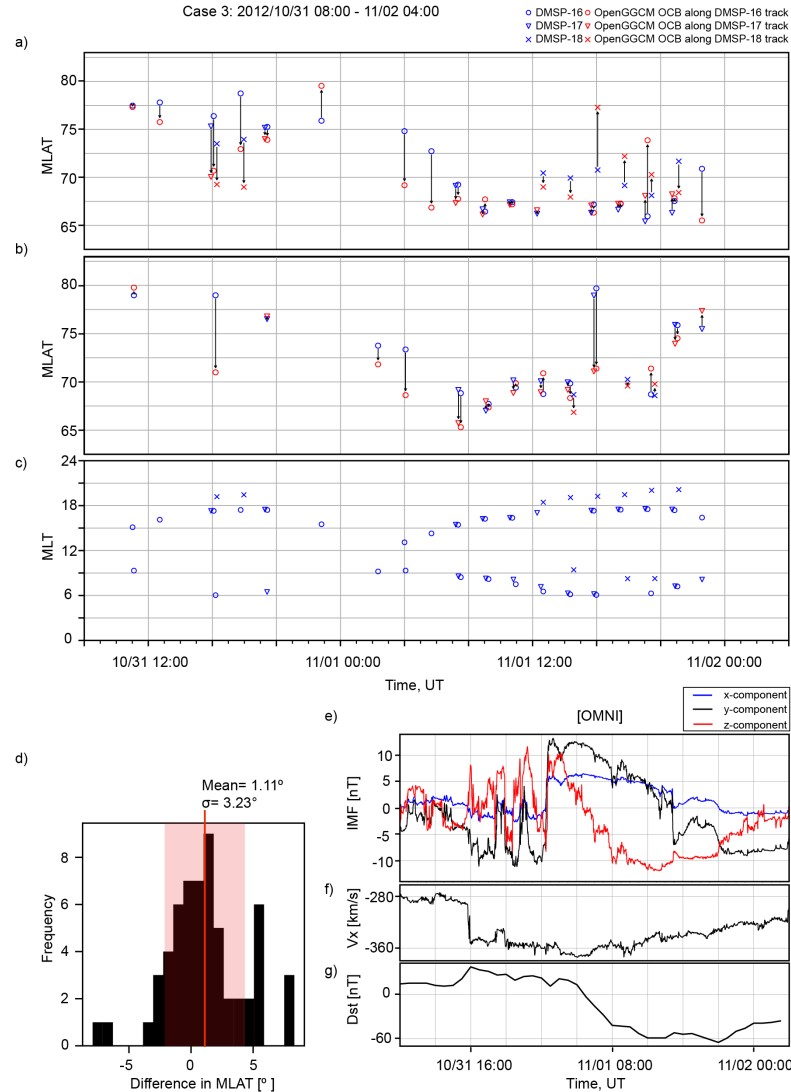

**Figure 3.** Comparison between the observed and simulated OCB latitudes for November 1, 2012 event in the same format as Fig. 1.

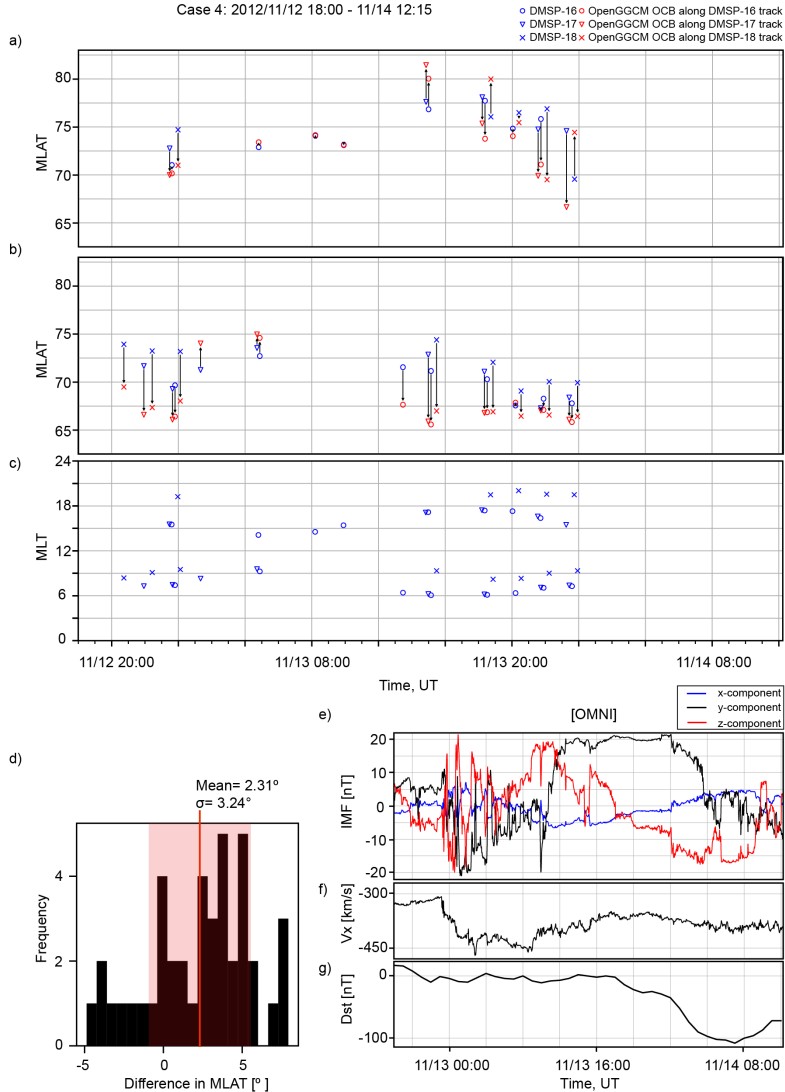

**Figure 4.** Comparison between the observed and simulated OCB latitudes for November 13, 2012 event in the same format as Fig. 1.

10, 2018 solar event shown in Fig. 5 has the largest average difference between the modeled and observed OCB latitudes, i.e., $4.12 \pm 2.20°$. However, this event also has by far the fewest data available.

To summarize the model-data comparisons, on average, the model overestimates the location of the polar cap boundary by $1.61°$ in latitude across all five storm events studied here. Thus, the model appears to be biased such that it predicts a larger polar cap than estimated from DMSP data. Overall, we consider this a satisfying result. In particular, each of the five events that we modeled lasted more than 24 hours. A similar study by Wang et al. (2018) showed on average a $2.33°$ deviation in OCB latitude between DMSP and PPMLR-MHD model for a substorm event that lasted about 10 hours. The reason for the positive

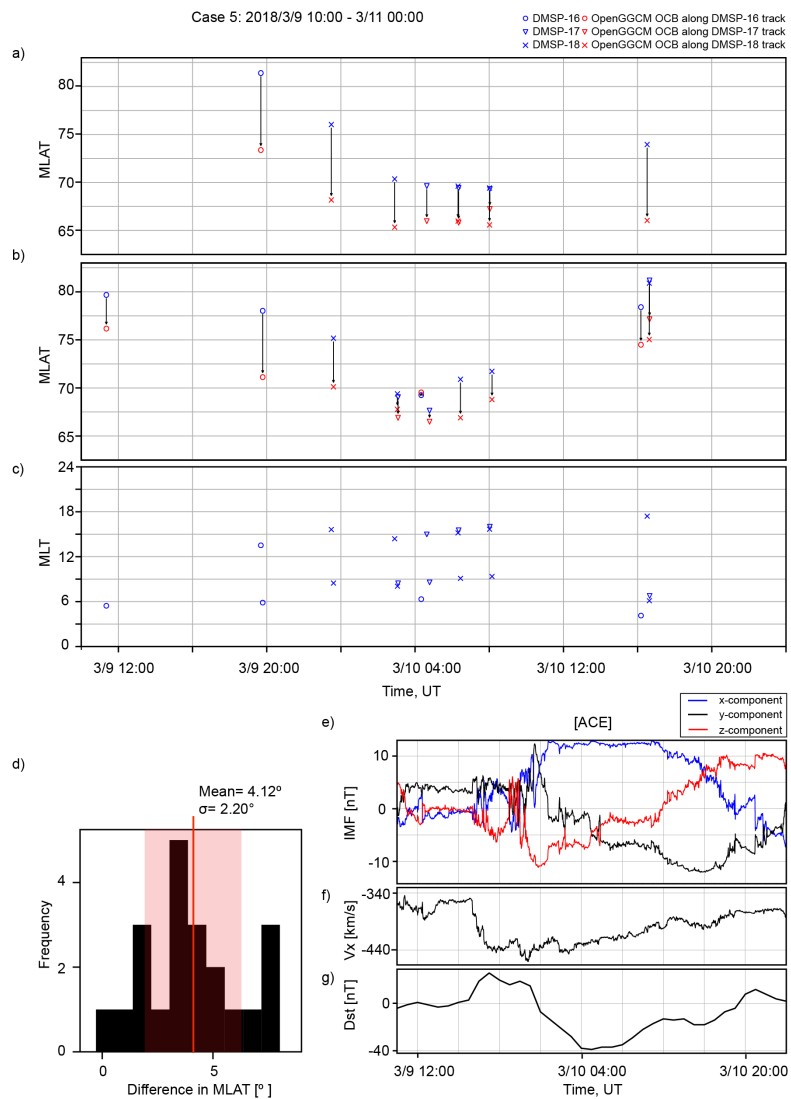

**Figure 5.** Comparison between the observed and simulated OCB latitudes for March 10, 2018 event in the same format as Fig. 1.

bias is not obvious. One might hope that there is a pattern in the solar wind or the IMF that might give a hint, but nothing stands out, although there is a trend that the larger storms seem to give more accurate model results than the weaker ones. However, the number of storms studied is too small to allow for any compelling conclusions.

## 3.2 IMF clock-angle dependence

As demonstrated in Fig. 1b through 5b, the DMSP coverage is not sufficient to provide a complete 24-hour MLT picture of the OCB dynamic during a geomagnetic storm. To assess this dynamic in more detail, the complete polar cap boundary from the

OpenGGCM model is plotted for the duration of four solar storm events in the top graphs in each panel of Fig. 6. The color-coded maps represent the OCB latitude as a function of time and MLT in a keogram style plot. This type of graph most clearly shows the formation and propagation of the polar cap expansion when the solar storm reaches the Earth and the IMF changes strength and direction. To investigate the temporal evolution of the expansion, we superimpose the IMF clock angle over the OCB maps. The lines representing the clock angle are shown in magenta to make them stand out from the color map. The IMF clock-angle is defined in such a way that when facing the Sun fully southward IMF has a clock -angle of $180°$, and when IMF $B_y > 0$, $B_z = 0$, the clock-angle is $90°$. The lower sub-panels in Fig. 6 a-c show the IMF $B_y$ and $B_z$ components, which are ultimately driving the polar cap expansion. From Fig. 6 it is readily apparent that there is a strong correlation between the IMF clock angle and the MLT of the largest polar cap expansion.

Figure 6a shows the polar cap dynamics for the October 1, 2012 event. The polar cap expands rapidly as the IMF turns southward at around 14:00 UT. That expansion first occurs at the dayside and the flanks. This expansion is coincident with the southward turning of the IMF. The expansion intensifies around 22:30 UT when the IMF southward component becomes stronger. As the IMF turns more and more northward, the polar cap shrinks, first at the flanks and then near noon. The magenta trace, i.e., the IMF clock angle, closely follows the red areas of the most expanded parts of the polar cap. While the polar cap is typically an oval that is displaced towards midnight, during storms like these, that appears no longer to be true. In particular, the strong southward IMF driving shifts the maximum extension to the dayside and to the flanks, depending on the clock angle, i.e., depending on the relative strength of the IMF $B_y$ component versus the $B_z$ component.

Figure 6b displays the OCB dynamics for the November 1, 2012 event. During this event, the IMF clock angle varies very smoothly through a 270 rotation during an interval of almost one day. Again, the polar cap expansion starts with the southward turn of the IMF. However, this southward turn is accompanied by a strong IMF $B_y$ component. Therefore, the expansion does not start at noon, but rather near the dawn terminator. The expansion then rotates towards noon as the IMF become s more southward. At the end of the interval, when the IMF is essentially duskward, the maximum of the expansion is near the dusk terminator. We also note that during the CME sheath phase the IMF is mostly northward, and correspondingly, there is no significant polar cap expansion.

The next event, shown in Fig. 6c in the same format, has a sheath that includes several significant southward IMF excursions. During each of these excursions, the polar cap rapidly expands. Like in the other cases, the direction of the expansion is dictated by the clock angle. First, the IMF is dawnward, and the expansion is towards dawn, followed by a duskward turn that is matched by the excursions. After the initial part of the sheath with southward IMF the remainder of the sheath has northward field, and the polar cap contracts. At $\approx$ 15:00 UT the IMF turns southward again, and the polar cap expands again. During the next $\approx$ 16 hours (tick marks are every 4 hours) the IMF $B_y$ component remains strong and positive, while the IMF turns increasingly south. This is different from the previous case, where the clock angle mostly changed because $B_y$ changed. However, just like in the previous case, the apex of the OCB follows the clock angle. This shows that the clock angle is the controlling factor, not the IMF $B_y$ component alone.

The next case, 10 March 2018, shown in Fig. 6d, is different in that the IMF slowly rotates from south to north, while IMF $B_y$ stays roughly constant and negative. The behavior of the OCB apex is as expected from the previous cases. As the clock

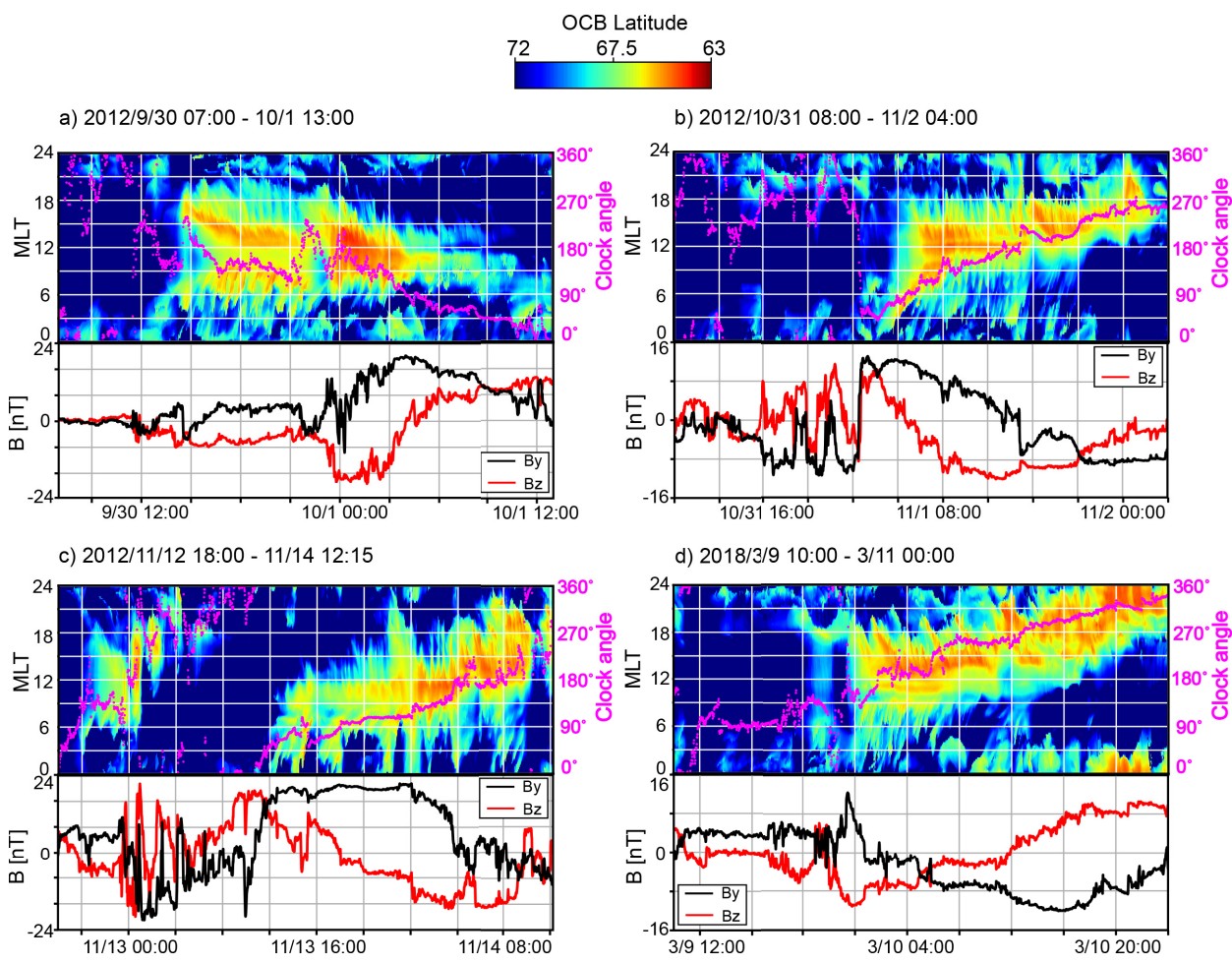

**Figure 6.** Modeled OCB dynamics during the four different geomagnetic storms. The color coding indicates the OCB latitude . Superimposed lines in magenta indicate the IMF clock-angle, with the scale shown on the right side of each panel. The bottom sub-panels show the $B_y$ (in black) and $B_z$ (in red) components of the IMF in GSE coordinates.

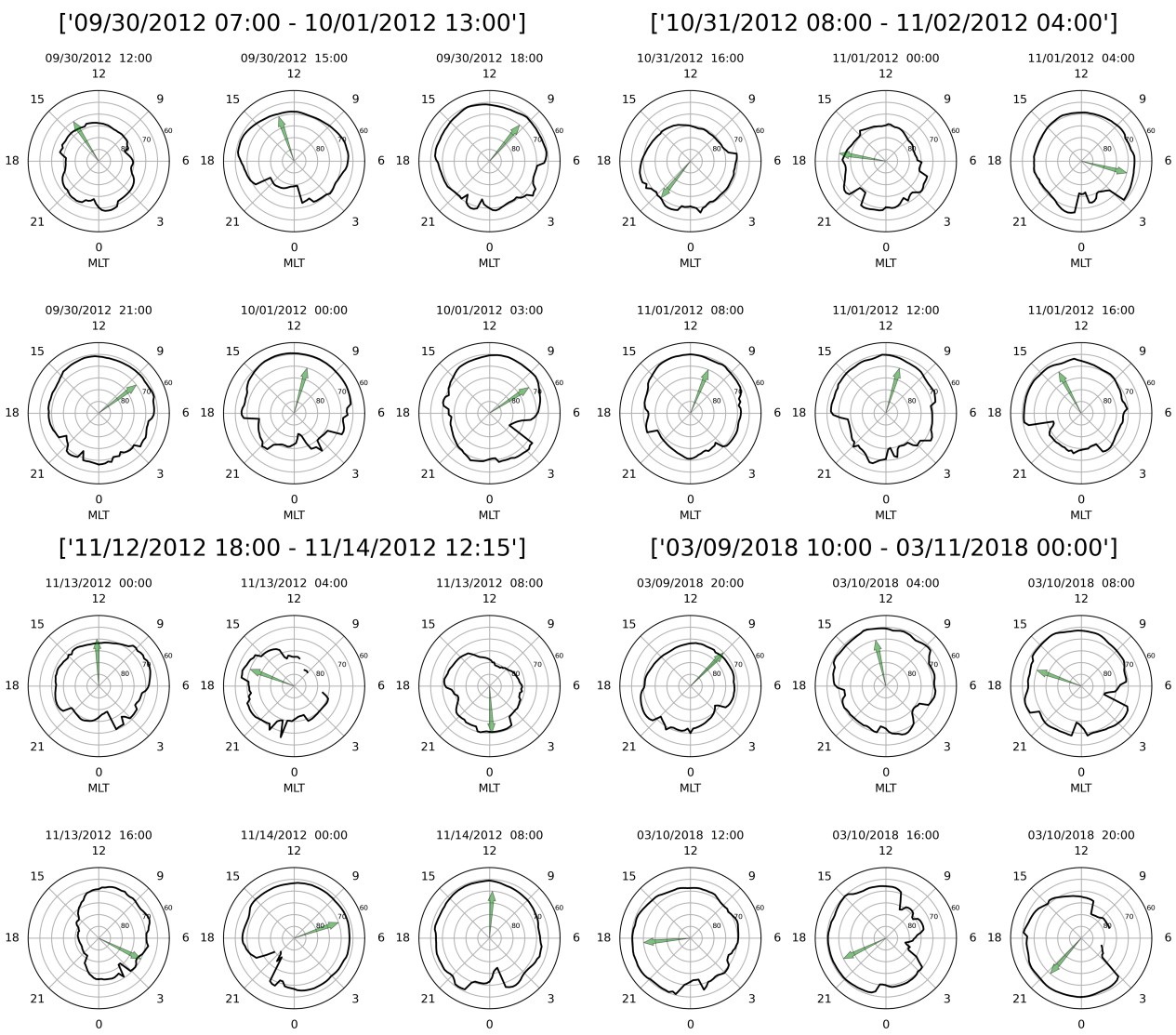

**Figure 7.** Four groups of polar plots corresponding to the clock angle and OCB latitude plots of Fig. 6. Within each of the groups the six polar views show the location of the OCB as a solid black line. The green arrows show the direction of the IMF vector as viewed towards the sun. It is evident that the maximum equatorward extent of the polar cap follows the clock angle direction of the IMF.

angle slowly rotates southward the OCB first expands near noon and then the apex rotates towards dusk following the clock

angle. A significant difference compared to the previous cases is that the rotation continues all the way to midnight. That shift of the apex to midnight occurs while the IMF is nearly northward.

Figure 7 provides a different perspective on the polar cap expansion. For the first four of the cases presented above, the figure shows a group of six polar views of the OCB at selected times. The green arrows show the IMF clock angle at the same time.

The figure confirms the previous findings, i.e., that the polar cap expansion closely follows the clock angle. The figure also shows that in most cases the OCB is fairly smooth where it is most expanded. When the OCB contracts, it tends to become a more ragged line. The latter may be explained from the fact that tail reconnection is localized, and thus the poleward motion of the OCB becomes localized as well.

Finally, Fig. 8a shows the strongest storm considered here, which occurred on 19-21 November 2003. This storm also has the prototypical full circle IMF rotation of a flux rope. As in the other cases, the OCB apex follows the clock angle. This occurs not only through the storm main phase, but also during the sheath rotation and the two other rotations that follow the storm main phase. We will later use this case for numerical experiments to rule out other causes than the clock angle for the polar cap expansion behavior.

### 3.3 Effects of flipping $B_y$ and $B_z$ components for the November 20, 2003 CME

In order to test whether the clock angle is the agent solely responsible for the asymmetric polar cap expansion, we conduct three numerical experiments by flipping the signs of the IMF $B_y$ and $B_z$ while keeping all other parameters the same. In Fig. 8b both signs are flipped, in 8c only the sign of the IMF $B_y$ component is flipped, and in Fig. 8d only the sign of IMF $B_z$ is flipped.

By doing so, we try to exclude other possible influences on the OCB, for example, seasonal effects such as dipole tilt (Russell et al., 2003) and ionosphere conductance distribution (Lu et al., 1994). Also, the clock angle changes occur in different directions. Specifically, between cases 8a and 8c, and between 8b and 8d the clock angles are reversed. It is obvious that the clock angle alone is the controlling factor. In particular, Figures 8a and 8c, and 8b and 8d, respectively, are virtual mirror images of each other. There is some net shift of the OCB in all of these cases, which is likely due to season, since the event date is close to winter solstice.

The experiment also tells us that the direction in which a clock angle change occurs is not very important. That, in turn, also means that the reaction to clock angle changes occur with only small time delays, at least on the time scales of these plots. Closer inspection of the color-coded plots show distinct streaks that indicate how a polar cap (PC) expansion propagates along the magnetopause. Since the time it takes for an IMF change to pass over the magnetosphere is of the order of 30 minutes, these features are washed out on the plots presented here. These details will be the subject of subsequent studies.

### 3.4 Further model-data comparisons

The key 'product' of our investigation is a database of DMSP crossings of the OCB with date, magnetic latitude (MLAT), and MLT of the crossing, tagged with the IMF clock angle (CLK) at that time, and the crossing MLAT from the model. This database has 297 entries and is part of the paper as supplemental material. This database allows us to further support our findings.

First, we present in Figure 9 a scatter plot to show the correlation between the model and the DMSP data. If the correlation was ideal all points would lie on the green line. The blue line is a linear least square fit. As we found before, there is substantial scatter, however, the model follows by-and-large the data.

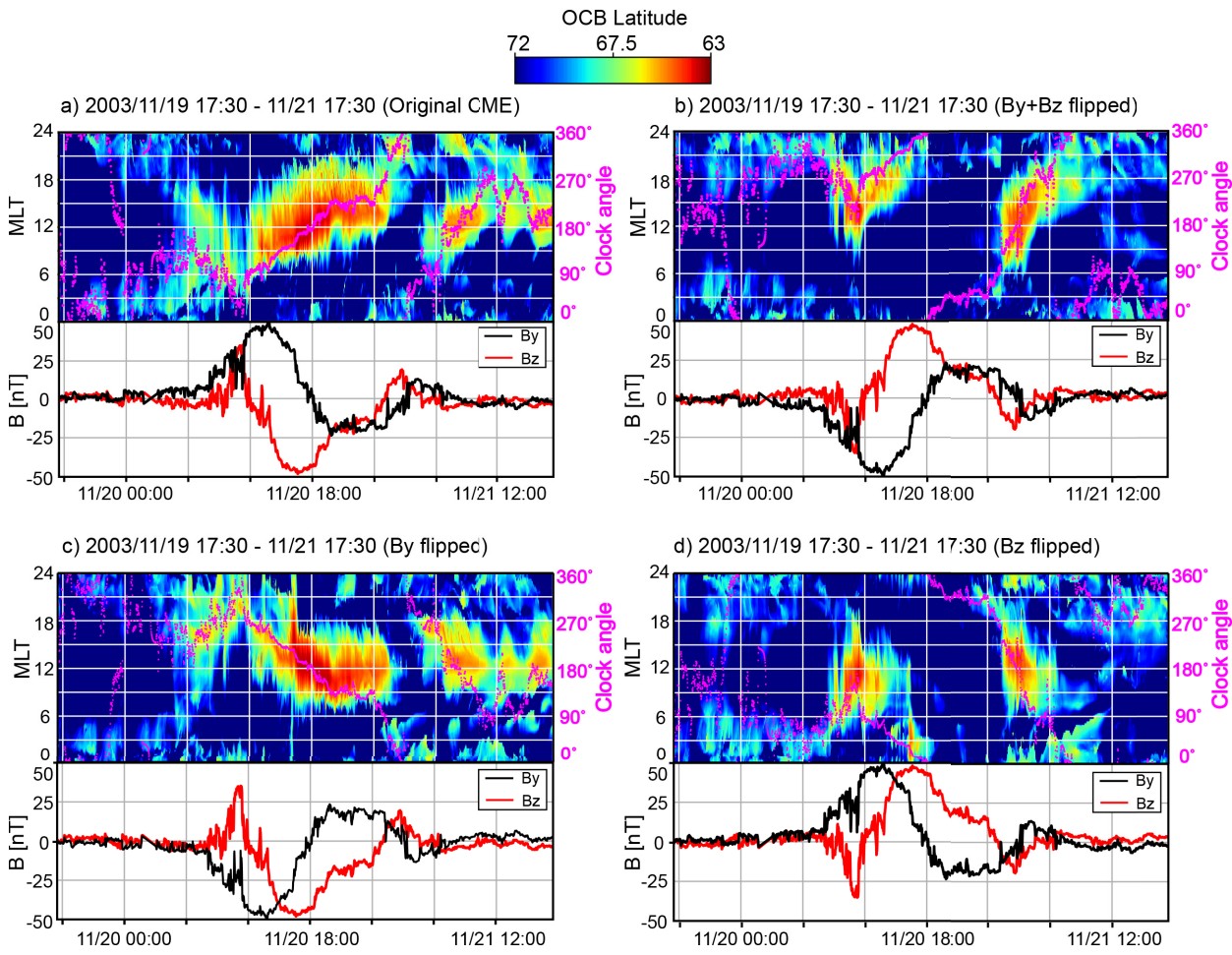

**Figure 8.** Modeled OCB dynamics during the November 20, 2003 geomagnetic storm. Color-coded colormaps indicate the OCB latitude. Superimposed plots in magenta indicate the IMF clock-angle. Bottom plots in each panel show the $B_y$ (in black) and $B_z$ (in red) components in GSE coordinates. Panel a) shows the original CME. Panel b) is the run with both $B_y$ and $B_z$ flipped. Panels c) and d) display cases where $B_y$ and $B_z$ were flipped correspondingly.

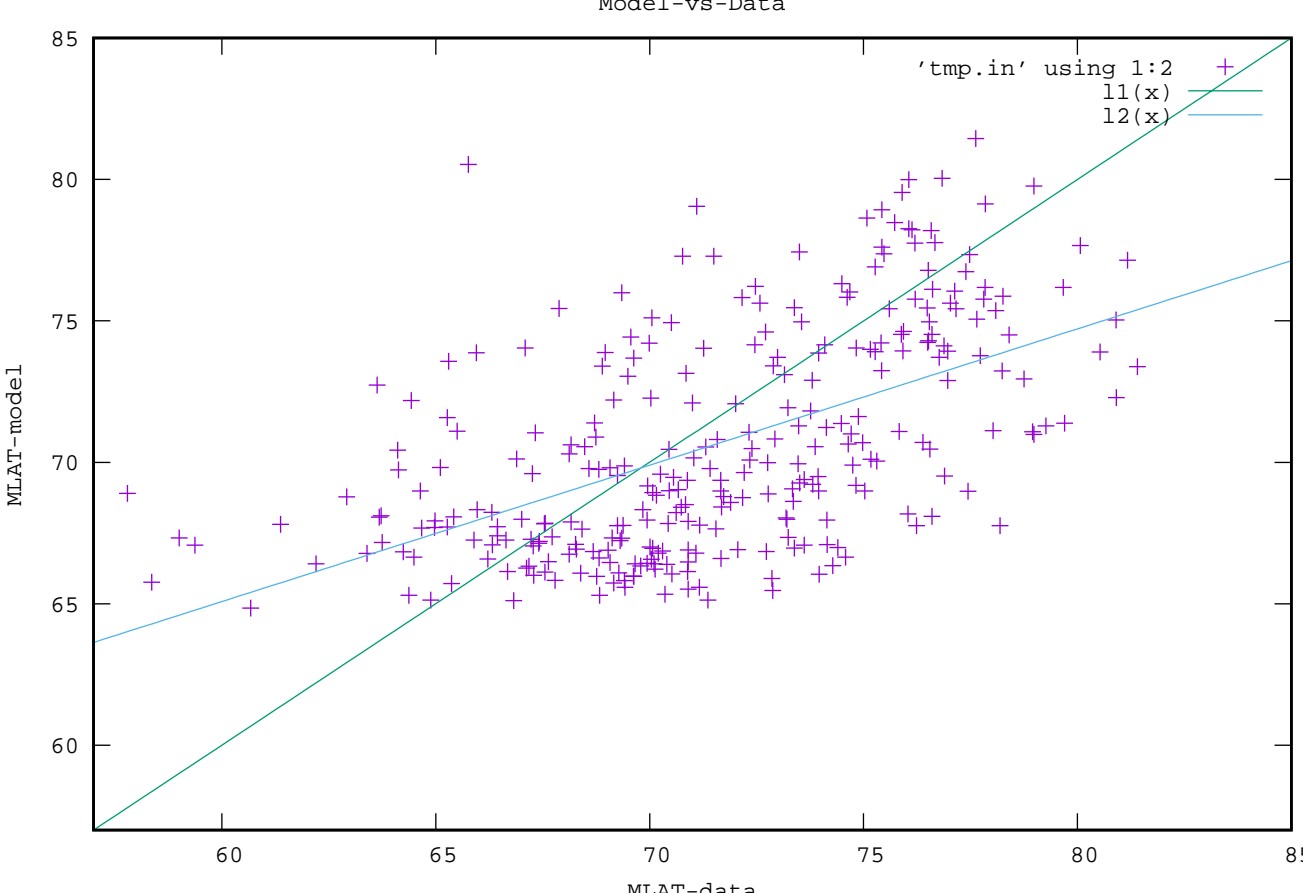

**Figure 9.** Correlation between the OCB latitudes obtained from DMSP and from OpenGGCM. The blue line is a linear fit, and the green line represents an ideal correlation.

The model never produces an OCB lower than about 64 degrees MLAT, while DMSP observes some crossings at much lower latitudes, down to less than 60 degrees. That is obviously a deficiency of the model and explains the shallower slope of the correlation.

Second, we further explore the expansion as a function of the IMF clock angle.

Figures 10 and 11 are scatterplots of the OCB latitude versus the IMF clock angle CLK, for the data and the model, respectively. In both plots a clock angle dependence is obviously such that the lowest latitudes correspond to southward IMF, which is expected. There is large scatter both in the data and in the model results. This is also expected, because other parameters, such as IMF magnitude, solar wind density, and solar wind velocity, also affect the expansion and are not taken into account

here. In order to make a quantitative comparison we fitted a cosine to each of the scatterplots. A cosine fit is a natural choice

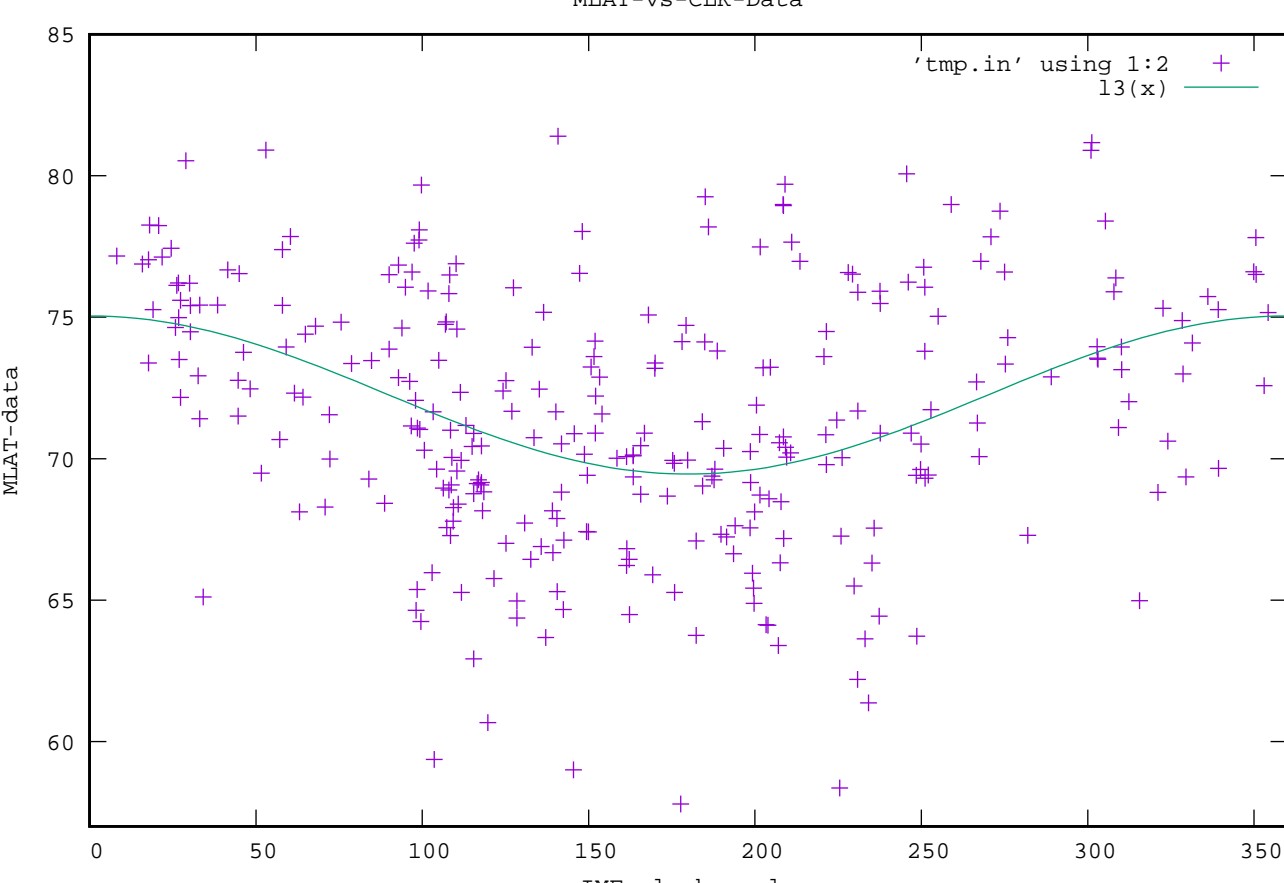

**Figure 10.** Scatterplot of MLAT versus IMF clock angle for the DMSP data. The green curve is a fit to the data with a simple cosine function.

because the clock angle is periodic and the distribution has a minimum at 180 degrees. In spite of the large scatter, the fits are very similar. The model is biased to lower latitudes by about 1 degree, i.e., 74 degrees versus 75 degrees maximum and 69 degrees versus 70 degrees minimum.

It is not possible (or useful) to create keogram-like plots of MLAT versus time and MLT as we did for the model results, because the data are too sparse. On the other hand, the database should contain this information, but a proper visualization needs to be applied. We proceed as follows: For both the data and the model we bin the average MLAT as a function of MLT and IMF clock angle (CLK). Since the data are sparse and there are some MLT/CLK combinations that have no data, primarily near noon and near midnight, because of the DMSP orbits. We display the grid with colored dots according to the mean latitude of the crossings in a given MLT/CLK bin. We choose 12 bins in each variable. Finer bins may reveal more structure but also produce more empty bins, but our choice is sufficient to support our conclusions. Figures 12 and 13 show the result. For both

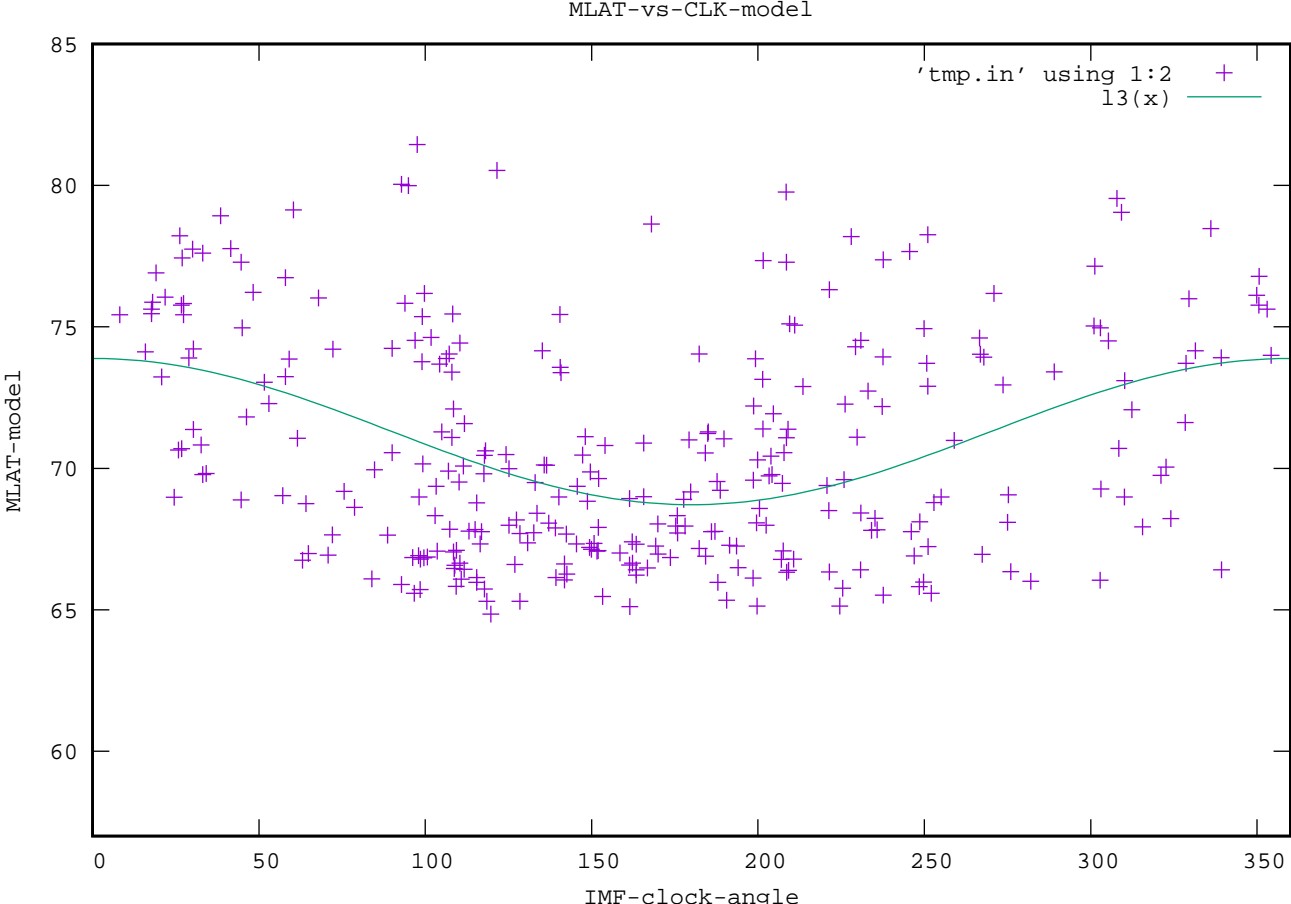

**Figure 11.** Scatterplot of MLAT versus IMF clock angle for the OpenGGCM results. The green curve is a fit to the data with a simple cosine function.

the model and for the data there is a clear pattern such that the lowest MLAT (largest polar cap expansion, light yellow) at a given MLT follows the clock angle CLK, or, vice versa, for a given CLK the maximum expansion occurs at a specific MLT, and the result is essentially identical for the model and the data.

## 4    Summary and conclusions

A comparison between the observed and modeled OCB locations along the DMSP satellites' paths for the five geomagnetic storm events was performed. Measurements from the DMSP particle instruments provided an estimated location of the polar cap. Using the orbit data, OpenGGCM model output for the selected events was extrapolated and the OCB latitudes were calculated. The comparison shows that on average the model overestimates the OCB latitude by 1.61° across all five solar

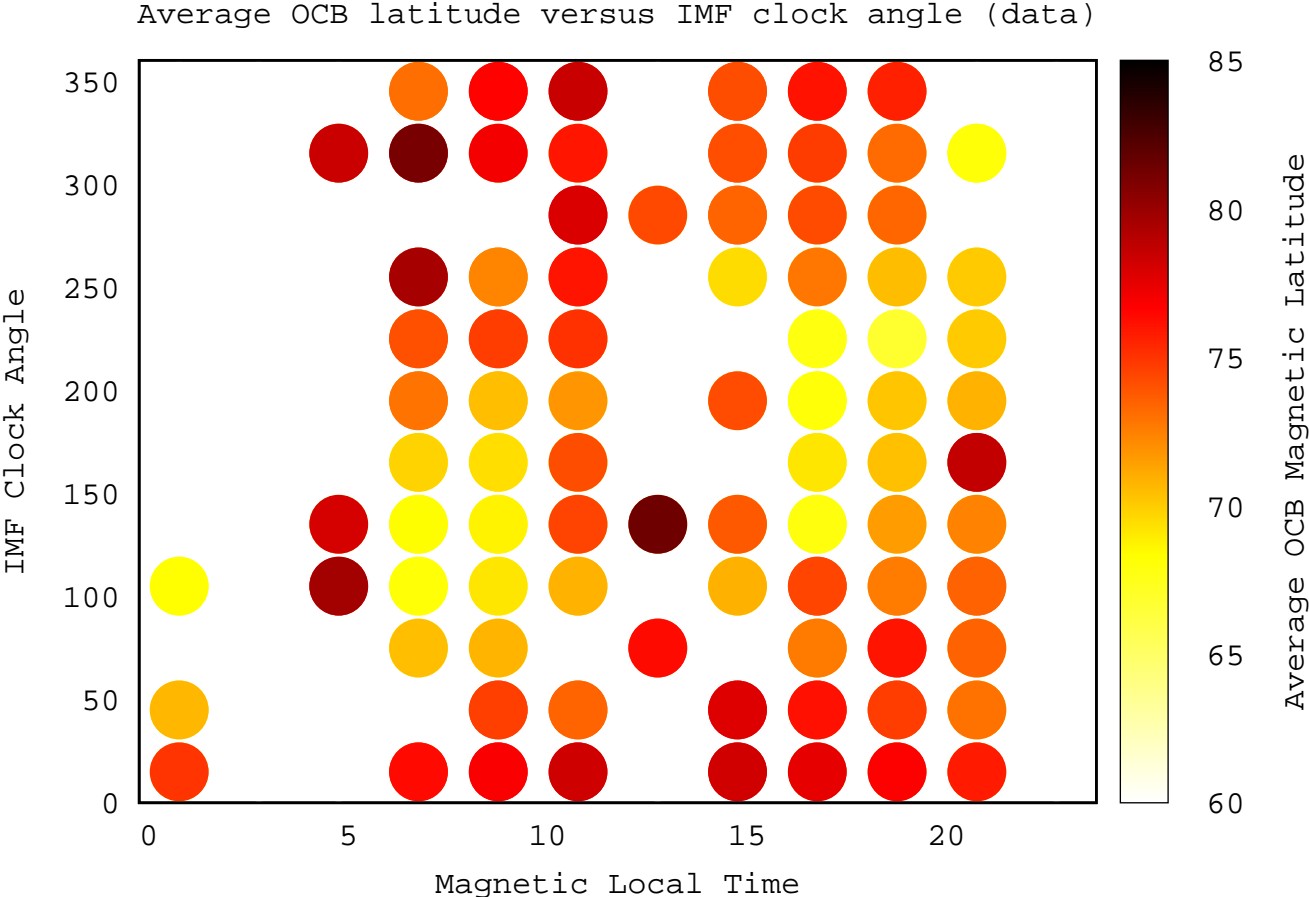

**Figure 12.** Average MLAT (over all events)as a function of MLT and CLK for the DMSP data.

storm events. This result is compelling in comparison to other studies and models performed for quiet time events, and shorter,
less intense substorm events (Rae, 2004; Wang et al., 2016, 2018).

When the IMF clock-angle trace is superimposed over the OCB colormap, it is readily obvious that the expansion of the
OCB is strongly correlated with the IMF clock-angle. However, this is only so when the IMF has a southward orientation.
During times of northward IMF there is no apparent correlation other than that the OCB moves to higher latitude, consistent
with previous studies (Newell et al., 1996). However, our particular choice of the colormap may also obscure some more
subtle details of the OCB during northward IMF periods. As soon as the IMF has a southward component, the PC rapidly
expands. This is consistent with the onset of magnetic reconnection at the dayside magnetopause. As reconnection begins on
the dayside, the PC first expands at the MLT of the reconnection site. The latter is controlled by the IMF clock angle (Pu et al.,
2007; Fuselier et al., 2010; Dunlop et al., 2011; Petrinec et al., 2016; Trattner et al., 2007, 2021), which was also demonstrated

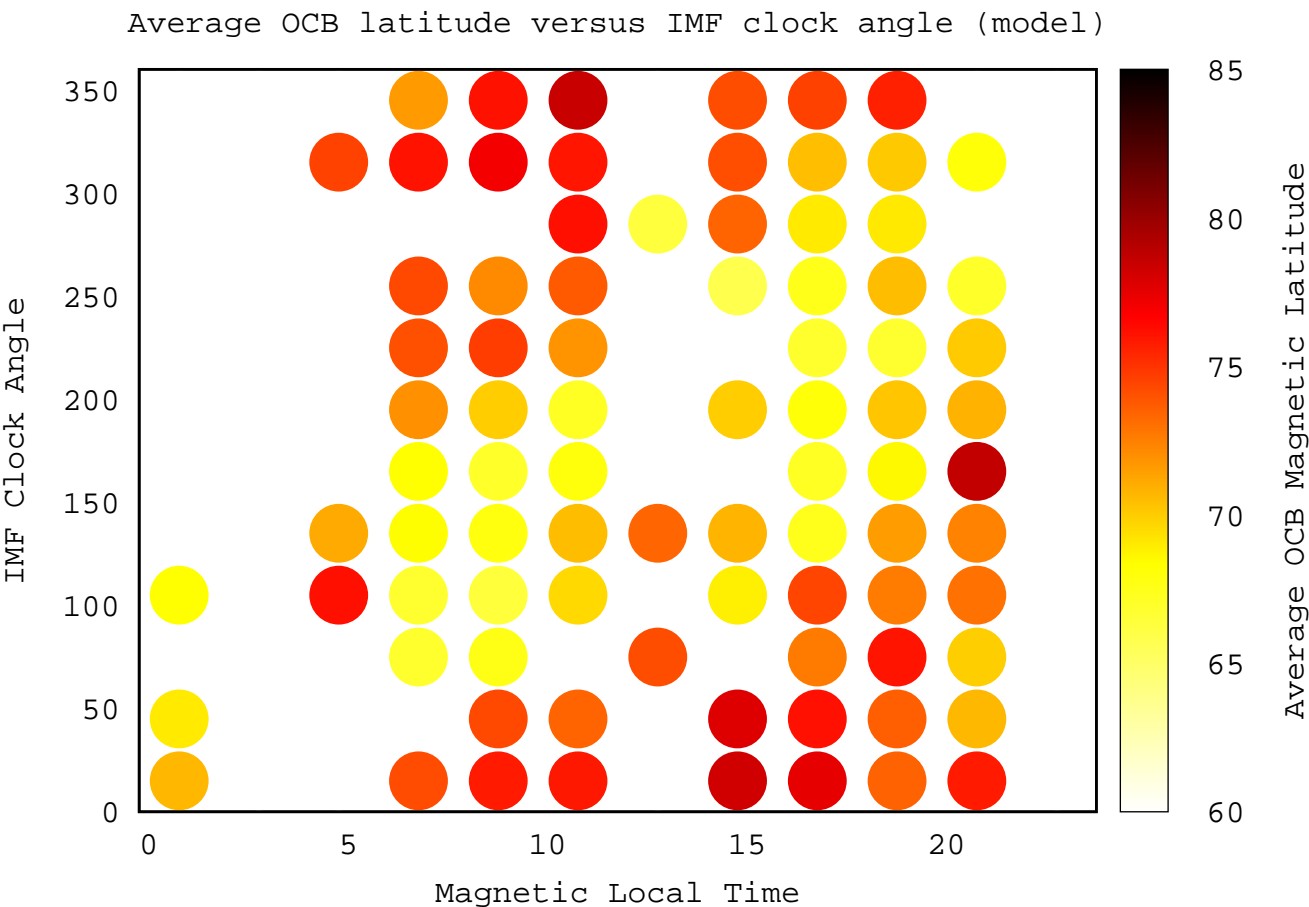

**Figure 13.** Average MLAT (over all events)as a function of MLT and CLK for the OpenGGCM results.

by auroral observations (Frey, 2003; Phan et al., 2003). This explains why a dawnward IMF opens up the polar cap first at
dawn and a duskward IMF at dusk.

It has been shown that the reconnection location responds very rapidly to clock angle changes (Trattner et al., 2007; Frey,
2003). Figures 6 and 8 show that the changes of the PC expansion are not as rapid. Once the PC has expanded in a particular
sector, it stays like that for some time. Eventually, convection will redistribute the flux in the polar cap (Cowley, 1982), but that
occurs on the time scale of hours.

It is not clear if the whole polar ionosphere (convection pattern) immediately responds to the IMF variation or if it needs
some time for the IMF changes to propagate from the dayside cusp region to the nightside auroral oval through the polar cap.
Fear and Milan (2012) argued based on the formation and motion of the transpolar arcs that the convection of magnetic field
lines should take a number of hours from the dayside to the nightside.. Zhang et al. (2015) suggested that cross-cap transit
time of the field line is about 1-2 hours from noon to midnight in MLT and that the time scale is 3 hours for the convection of

the full Dungey cycle by tracing the polar cap patches. Browett et al. (2017) suggested that the timescale varied from 1 to 5 hours for the penetration of IMF $B_y$ into the magnetotail depending on the IMF orientation and solar wind speed. However, as soon as a closed field line reconnects it expands the polar cap, so that process is instantaneous. Still, convection redistributes open flux, so some delay can be expected. That may be the cause of the still significant scatter in the MLAT-CLK plots. On the other hand, the cases presented in this paper have for the most part prolonged intervals (several hours) of fairly constant IMF or slowly changing IMF, thus such time delays should have only a minor impact. Close inspection of the MLAT versus MLT/time figures shows streaks at a much smaller time scale ($\sim$30 min) that seem to be associated with polar cap reconfiguration. We will study this in more detail in forthcoming work.

Reconnection in the plasma sheet will eventually close the open flux again, and convection will bring it back to the dayside, closing the Dungey cycle (Milan et al., 2007). This normally occurs on the substorm time scale, i.e., 1-2 hours. However, during storms, strong magnetopause reconnection continues for much longer times, and thus the polar cap does not recover as quickly. Ultimately, a balance between dayside reconnection, night side reconnection, and convection needs to occur, which also determines when and how the PC expansion saturates.

In summary, we find that 1) the comparison between DMSP and OpenGGCM OCB locations show that the model predicts the general trends of OCB well but not individual crossings, 2) the apex of PC expansion follows the IMF clock angle during periods of flux rope rotation of the IMF, and 3) during times of strongest southward IMF the PC shifts towards the dayside.

*Data availability.* The solar wind and activity indices were provided by the NASA CDAWeb and OMNIWeb sites. The DMSP particle detectors were designed by Dave Hardy of AFRL, and data obtained from JHU/APL (http://sd-www.jhuapl.edu/Aurora/spectrogram/index. html) .The numerical results shown in figures 1-8 are available on Figshare.com (https://doi.org/10.6084/m9.figshare.14499348.v1).

*Author contributions.* BT performed the simulation runs and produced the graphical output. JR conceived the study and led the analysis. WDC contributed to code development, simulation runs, and analysis. BF, TJF, NM, and RJS contributed to the analysis of the data and simulation results and their interpretation.

*Competing interests.* The authors declare that they have no conflict of interest.

*Acknowledgements.* This work was supported by the Air Force Office of Scientific Research under award number FA9550-18-1-0483 and NASA grant 80NSSC18K1220. The authors wish to thank Rob Redmon and Steve Petrinec for help with the DMSP data. Computations were performed on Marvin, a Cray CS500 supercomputer at UNH supported by the NSF MRI program under grant AGS-1919310.

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
