# Peer review of "Storm time polar cap expansion: IMF clock angle dependence"

_Annales Geophysicae, 2022_

## Author Comment (AC1)

Response to referee #2:

We would like to thank the referee for his/her thoughtful comments and suggestions, which helped us greatly to improve the paper.  In the following we quote the referee's comments in italics, followed by our response. Annales handles the process a little bit different than most other journal such that they only ask for our response at this point, but not for a revised manuscript.  We therefore include new figures and text in the response and will include them in the final manuscript when the editor asks us to do so.

*This paper presents the locations and dynamics of polar cap boundary during the five magnetic storm periods by using OpenGGCM-CTIM-RCM simulation together with the DMSP observations. The polar cap boundary and area are crucial parameters for the energy features in the solar wind-magnetosphere coupling, which are strongly related to solar wind and interplanetary magnetic field conditions. Therefore, the topic is very important. This papar shows the simulation and observation results of IMF clock angle dependence of the polar cap boundary mainly are consistent with each other, eventhough the simulation overestimates the latitude of boundary. Moreover, it shows the MLT of the largest polar cap expansion closely correlates with the IMF clock angle and the simulation can remedies observation limitation in local time. These results give new insights into the dynamics of polar cap expansion during storm time, and this paper is of course suitable to be publish in Annales Geophysicae. However, before it is published, I recommend the authors address the following points:*

*Major comments:*

   *In Lines 91-92, the authors mentioned "the coupled numerical model is driven by observed solar wind and IMF data at Lagrangian 1 (L1)" and "…are obtained from OMNIWeb". These seem to confuse the readers where the interplanetary data you used in your model are, L1 or nose of bow shock? Please make it clear in the text. Here I assume your input data have been shift to the nose of bow shock from OMNIWeb. If so, have you considered any time delay for the solar wind and IMF data propagating from the bow shock to the dayside magnetopause and the polar ionosphere? As we know the solar wind and IMF need several minutes to reach the dayside magnetopause by crossing the magnetosheath, and need 1-2 minutes to affect the dayside polar ionosphere when the solar wind-magnetosphere coupling happened at the dayside magnetopause. Thus, the time delay is very important when you compare the IMF with your observation and simulation results. I suggest the authors to consider these time delays at least for the delay for the solar wind and IMF crossing through the magnetosheath.*

The NASA web site for the OMNI data describes the time shifts in great detail, see section 12 in: https://omniweb.gsfc.nasa.gov/html/ow_data.html.  In particular, the data are shifted (by ballistic propagation, which is the best one can do with the available information) to 30 RE upstream of Earth.  Although OpenGGCM has its own internal algorithm for time-shifting solar wind inputs (similar to the OMNI procedure) it also has the inflow boundary at 30 RE on the sunward side, so no further time shift is necessary.

   *I remember that there is a debate for a long time: whether the whole polar ionosphere (convection pattern) immediately responses the IMF variation or need some time for the IMF effect propagating from dayside cusp region to nightside auroral oval through the polar cap. Fear and Milan (2012) argued the convection of magnetic field lines should take a number of hours from the dayside to the nightside after statistically analyzed the formation of the transpolar arcs. Zhang et al. (2015) suggested that cross-cap transit time of the field line is about 1-2 h from noon to midnight in MLT and is 3 h for the convection of the full Dungey cycle by tracing the polar cap patches. Browett et al. (2017) suggested that the timescale varied from 1 to 5 h for the penetration of IMF By into the magnetotail depending on the IMF orientation and solar speed. I think the OCB in different MLT may need different time to response the IMF variations. Thus, we may need to be careful when we compare the OCB at all MLT with the IMF clock angle, such as in Figure 6. I suggest the authors make any clarification of these in the text, because this is related to your main results.*

We agree that the convection across the polar cap takes some time. However, as soon as a closed field line reconnects it expands the polar cap, so that process is instantaneous. Still, convection redistributes open flux, so some delay can be expected. That may be the cause of the still significant scatter in the MLAT-CLK plots (see our response to the other reviewer.) We will add some discussion addressing this point and in particular refer to the references. On the other hand, the cases presented in this paper have for the most part prolonged intervals (several hours) of fairly constant IMF or slowly changing IMF, thus such time delays should have only a minor impact. Close inspection of the MLAT versus MLT/time figures shows streaks at a much smaller time scale (~30 min) that seem to be associated with polar cap reconfiguration. We will study this in more detail in forthcoming work.

*Small suggestions and typos:*

   *How do the authors identify the OCB from the DMSP spectrograms? Although the authors cited some references, I suggest to make a briefly interpretation about it for the readers easily following, because the OCB is the main topic of this manuscript.*

We addressed this point in the response to reviewer #1.

   *Line 130: "in Fig. 1 through ??" Please check.*
   *Page 6, the caption of Figure 1 line 1: There is an extra space in DMSP.*
   *Lines 186-190: These sentences are not clear to me. Could you please explain how the northward IMF influence on the behavior of OCB?*
   *Line 215: PC-> polar cap (PC). When an abbreviation of term first appears in the manuscript, the term should have a full name first.*
   *Lines 219-220: please tone down of the express "five strong geomagnetic storm events", because there is an event with a minima Dst of -39 nT, and an event with a minima Dst of -65 nT in your events list.*

We appreciate the suggestions and will rephrase those statements in the final manuscript accordingly.

*References mentioned above:*

*Browett, S. D., R. C. Fear, A. Grocott, and S. E. Milan (2017), Timescales for the penetration of IMF By into the Earth's magnetotail, J. Geophys. Res. Space Physics, 122, 579–593, doi:10.1002/2016JA023198.*

*Fear, R. C., and S. E. Milan (2012), The IMF dependence of the local time of transpolar arcs: Implications for formation mechanism, J. Geophys. Res., 117, A03213, doi:10.1029/2011JA017209.*

*Zhang, Q.-H., M. Lockwood, J. C. Foster, S.-R. Zhang, B.-C. Zhang, I. W. McCrea, J. Moen, M. Lester, and J. M. Ruohoniemi (2015), Direct observations of the full Dungey convection cycle in the polar ionosphere for southward interplanetary magnetic field conditions, J. Geophys. Res. Space Physics, 120, 4519–4530, doi:10.1002/2015JA021172.*

*The end.*

---

## Author Comment (AC2)

Response to referee #1:

We would like to thank the referee for his/her thoughtful comments and suggestions, which helped us greatly to improve the paper. In the following we quote the referee's comments in italics, followed by our response. Annales handles the process a little bit different than most other journal such that they only ask for our response at this point, but not for a revised manuscript. We therefore include new figures and text in the response and will include them in the final manuscript when the editor asks us to do so.

*This paper attempts to show that:*

*Open GGCM used in the paper does a good job of predicting the location of the open-closed magnetic field boundary (OCB)*
*The minimum latitude reached by the polar cap expansion during a storm follows the IMF clock angle during periods of rotation of the IMF*
*During times of strongest southward IMF, the polar shifts towards the dayside.*

*Disappointingly, in my opinion, the paper does not demonstrate these points.*

*The claim that that the model does a good job of predicting the OCB is not supported by the data shown in the paper. The paper shows histograms of the error of the predicted OCB compared to the observations for each of four storms, and it is readily seen that the histograms and standard deviation are roughly what would be expected for a uniform distribution of error over the range +/-5 degrees in latitude. This indicates to me that the model is essentially giving a random location for the OCB over a 10 degree range of latitudes, a range that is larger than the typical width of the auroral oval. There is some trend for both the model and the data to show the well-known tendency for the OCB to move to lower latitudes with increasingly southward IMF Bz.*

This criticism is well taken. We simply had not done a good job examining and displaying out results. The key 'product' of our investigation is a database of DMSP crossing of the OCB with date, magnetic latitude (MLAT), and MLT of the crossing, tagged with the IMF clock angle (CLK) at that time and the crossing MLAT from the model. This database has 297 entries and will be added to the paper as supplemental material. The database contains all the information necessary to address the referees' points, but requires a proper display. We will add the following figures to the paper (they are labelled consecutively here but will get the appropriate numbering when included in the manuscript):

[Figure]

Figure 1. A simple correlation between the OCB latitudes obtained from DMSP and from OpenGGCM. The blue line is a linear fit, and the green line represents an ideal correlation.

Figure 1 shows that the model and the data indeed correlate, albeit with a large scatter. The model never produces an OCB lower that about 64 degrees MLAT, while DMSP sees some crossing at much lower latitudes, down to less that 60 degrees. That is obviously a deficiency of the model and explains shallower slope of the correlation.

Figures 2 and 3 are scatterplots of the OCB latitude versus the IMF clock angle CLK, for the data and the model, respectively. In both plots a clock angle dependence is obvious such that the lowest latitudes correspond to southward IMF. There is large scatter both in the data and in the model results. In order to make a quantitative comparison we fitted a cosine to each of the scatterplots. A cosine fit is a natural choice because the clock angle is periodic and the distribution has a minimum at 180 degrees. In spite of the large scatter, the fits are very similar. The model is biased to lower latitudes by about 1 degree, i.e., 74 degrees versus 75 degrees maximum and 69 degrees versus 70 degrees minimum.

[Figure]

Figure 2. Scatterplot of MLAT versus IMF clock angle for the data. The green curve is a fit to the data with a simple cosine function.

[Figure]

Figure 3. Scatterplot of MLAT versus IMF clock angle for the model. The green curve is a fit to the data with a simple cosine function.

*The two conclusions about the polar cap location and shape are obtained solely from the model and no attempt is made to consider whether these trends are also seen in the data. It would be interesting if such trends could also be discerned from the data, and perhaps the authors would consider whether they are able make such a determination.*

It is not possible (or useful) to create plots of MLAT versus time and MLT like we did for the model results, because the data are too sparse. On the other hand, the database should contain this information, but a proper visualization needed to be found. We proceed as follows: For both the data and the model we bin and average MLAT as a function of MLT and IMF clock angle (CLK). Since the data are sparse and there are MLT/CLK combinations the have no data (primarily near noon and near midnight because of the DMSP orbits) we displayed the grid with colored dots according to the mean latitude of the crossings in a given MLT/CLK bin. We choose 12 bins in each variable. Finer bins may reveal more structure but also produce more empty bins, but our choice is sufficient to support our conclusions. Figures 4 and 5 show the result. For both the model (we knew that already) and for the data there is a clear pattern such that the lowest MLAT (largest polar cap expansion) at a given MLT follows the clock angle CLK, or, vice versa, for a given CLK the maximum expansion occurs at a specific MLT, and the result is essentially identical for the model and the data. We believe this should take care of the reviewer's concern that the data would not support our original conclusions.

[Figure]

Figure 4.  Average MLAT as a function of MLT and CLK for the model.  This is essentially just a different presentation of the relation averaged of all cases as were already presented in figures xx-yy.

[Figure]

Figure 5.  Average MLAT as a function of MLT and CLK for the DMSP data.  The pattern follows closely the one seen in the model results of Figure 4.

*A further point is that the paper does not adequately describe how the OCB was determined from the DMSP data. Many times the DMSP data shows a clean transition from the plasma sheet to polar rain, but this is far from always the case.  For example, low-energy (<~1 keV) can extend roughly continuously from the plasma sheet to the mantle on the morning side, and, in the vicinity of the cusp, the OCB can lie at an equatorward boundary of precipitation because cusp precipitation is on open field lines.  Enough information needs to be presented so that a knowledgeable person could reproduce the results, Simple saying "spectrograms of ion and electron differential*

*fluxes in a range from 30 eV to 30 keV were inspected to identify the polar cap boundary crossings of the satellites"*
*is not sufficient.*

We agree that identifying the OCB in the data is subjective and not an easy task. An alternative is to use an automated algorithm as the APL group has done, but when we checked those determinations by inspecting them against spectrograms we found them not reliable. We thus inspected each crossing individually and also in a very conservative manner. Crossings that could not be clearly identified are not included in the data base. Also, when we could identify polar cap precipitation features such as polar cap arcs, these were not included. We also never used data below ~1 keV, so there should be no concerns about the cusp. All crossings are tabulated in the data base with a time tag, so in the event a reader wants to double check he/she can do so.

*A minor point: I recommend not including statements of fact in a paper's Introduction without a reference.*
*Examples in the current paper are:*

*"Convection can also change the shape of the OCB without changing the flux contained in the polar cap."*

We will add the citation "M. Lockwood, S. W. H. Cowley, M. P. Freeman, The excitation of plasma convection in the high-latitude ionosphere, JGR, 1990, https://doi.org/10.1029/JA095iA06p07961"

*"When the polar cape opens up, that plasma leaves the plasmasphere and convects away. Thus, the OCB shape also controls the shape of the plasmasphere."*

We will add the citation: "A. Nishida, Plasmapause, Convection, and Reconnection, JGR, 2019, https://doi.org/10.1029/2019JA026898"

*"During times of high geomagnetic activity these methods can fail because the precipitation is very intense, clobbering the radars' return signal."*

This is a well-known fact and often mentioned, but apparently never put in writing, so we will remove this sentence.

---

## Author Response (AR1)

**Response statement addressing the upload of the revised manuscript:**

We are a little confused about the process.  We had previously responded to the reviewers and were told not to submit a revised manuscript at that time.  We were then later told to submit a revised manuscript, but there was no additional feedback from the reviewers, so we can only assume that they were satisfied with our response.  As we upload the revised manuscript, the web page asks us to submit a response as well, which we do with this statement.

Please let us know if we did anything wrong here.

Best regards – Jimmy Raeder. (jraeder9@gmail.com)

---

## Referee Report (RR1)

The authors have responded well to my comments and, in particular, have successfully shown the clock-angle effect in the DMSP data. I can recommend that the paper be published after some changes in the text to more accurately reflect the lack of agreement between the model and the data. As I said in my previous report "The paper shows histograms of the error of the predicted OCB compared to the observations for each of four storms, and it is readily seen that the histograms and standard deviation are roughly what would be expected for a uniform distribution of error over the range +/-5 degrees in latitude. This indicates to me that the model is essentially giving a random location for the OCB over a 10 degree range of latitudes, a range that is larger than the typical width of the auroral oval."

It is thus not accurate to say statements such as the statement in the Abstract "However, we generally find good agreement between the model and the observations.", and the statement at the end of the paper "the comparison between DMSP and OpenGGCM OCB locations show that the model predicts the OCB well". Please emphasize that the model does not do a good job of predicting the OCB at any particular time and location, but can reproduce the general trend as a function IMF Bz and clock angle.

---

## Author Response (AR2)

**Response statement addressing the upload of the revised manuscript:**

We made the requested minor modifications.

Best regards – Jimmy Raeder. (jraeder9@gmail.com)

---

## Author Response (AR3)

**Response statement addressing the upload of the revised manuscript:**

We made the requested minor modifications.

Best regards – Jimmy Raeder. (jraeder9@gmail.com)